# $t\bar{t}t\bar{t}$: NLO QCD corrections in production and decays for the $3\ell$ channel

**Nikolaos Dimitrakopoulos** ⋆

Institute for Theoretical Particle Physics and Cosmology, RWTH Aachen University, D-52056 Aachen, Germany

⋆ ndimitrak@physik.rwth-aachen.de

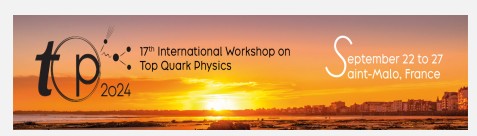

*The 17th International Workshop on Top Quark Physics (TOP2024) Saint-Malo, France, 22-27 September 2024* doi:10.21468/SciPostPhysProc.?

## Abstract

We discuss the results for the four-top quark production process at the LHC at NLO accuracy in perturbative QCD for the $3\ell$ decay channel. The QCD corrections are applied in both the production and the decay stages of the four top quarks by employing the narrow-width approximation. The spin correlations are therefore preserved at NLO accuracy in QCD without any approximation. We summarize the impact of higher-order QCD effects by highlighting the sensitivity of the results on the cut applied on the invariant mass of the two hardest light jets in the process.

P3H-24-090, TTK-24-51

## 1 Introduction

One of the rarest processes at the LHC is the simultaneous production of four top quarks. Despite its very small cross section, its study is of great interest and importance for several reasons. Firstly, it offers a unique opportunity to probe the top Yukawa coupling already at the Born level. Combined with the $t\bar{t}H$ production process, this allows for the derivation of exclusion limits on this coupling. Secondly, this process exhibits high sensitivity to various beyond the Standard Model (SM) scenarios, where new intermediate particles decaying into top-quark pairs could significantly alter the SM predictions. Lastly, it also serves as a direct probe for constraining the Wilson coefficients related to top-quark quartic couplings in the SM Effective Field Theory. Recently, the ATLAS and the CMS collaboration announced the discovery of four-top quark production by analyzing events with two same-sign, three, and four charged leptons [1, 2].

From the theory side, the first NLO QCD predictions for stable four-top production were obtained in Ref. [3] and later also in Ref. [4,5]. Besides the NLO QCD calculations, the inclusion of both QCD and EW corrections to all LO contributions was studied in Ref. [6]. Lastly, results at next-to-leading logarithmic accuracy were presented in Ref. [7], providing the most precise predictions to date for stable four-top production. However, for a more realistic study of the fiducial phase space, the inclusion of the top-quark decays is crucial and should also be considered. Towards that direction, in Ref. [8], results for the $pp \to t\bar{t}t\bar{t}$ process in the $1\ell + jets$ channel were matched to parton showers using the POWHEG framework. In this work, the NLO QCD corrections were applied only at the production stage of the top quarks, while the decays were modelled based on the method described in Ref. [9,10]. As a result, spin correlations were retained at LO accuracy in the soft and collinear regions and at NLO accuracy only for hard real emissions. Furthermore, QCD corrections to the top-quark decays were only described by the parton shower. To study the importance of the NLO QCD corrections to the top-quark decays, in Ref. [11] we presented results at NLO accuracy in perturbative QCD for the $4\ell$ channel using the narrow-width approximation (NWA). By doing so, the emission of the hard radiation was well described in the top-quark decays and spin correlations were preserved to NLO accuracy in the whole phase space without any approximation. In this report we briefly summarize the results obtained in Ref. [12] where a similar study was performed but for the more complicated $3\ell$ signal region. In section 2 we provide a description of our calculation, while the integrated and differential fiducial results are presented in Sections 3, 4 respectively. We finally conclude in Section 5.

## 2 Description of the calculation

All of the results provided in Ref. [12] were obtained using the NWA for the treatment of the top quarks and the $W$ bosons. This method allows us to separate the production and the decay stages and investigate the impact of the QCD corrections on each of these stages. Nevertheless, at NLO in QCD several NWA treatments are possible, and results are presented for three distinct scenarios:

- $NLO_{full}$: In this scenario, the QCD corrections are applied to both the production and decay stages, with the top-quark width treated as a fixed parameter calculated with NLO accuracy.

- $NLO_{exp}$ or simply $NLO$: The QCD corrections are again included in both stages. However, unlike $NLO_{full}$, an expansion in terms of the strong coupling constant is also applied to the top-quark width. This method serves as the default approach.

- $NLO_{LO_{dec}}$: The QCD corrections are solely applied to the production stage of the top quarks and the LO top-quark width is utilized everywhere in the calculation.

All of the aforementioned approaches are thoroughly discussed in Ref. [11]. The calculations are performed with the help of the HELAC-NLO Monte-Carlo framework [13] using a center of mass energy of $\sqrt{s} = 13.6$ TeV. All events are characterized by the presence of at least 4 b-jets, at least 2 light jets, exactly 3 charged fermions (electrons and muons) and missing energy associated with the presence of neutrinos in the final state. At NLO in QCD, the emission of an additional light jet can lead to situations where the two light jets from the $W$ boson are recombined into a single jet. In such cases, if the additional light jet is resolved and passes all selection cuts, it could act as the second resolved light jet, resulting in the event being accepted. However, previous studies (e.g., Ref. [14–16]) have shown that such configurations contribute to very large $\mathcal{K} = \sigma_{NLO}/\sigma_{LO}$ factors, which can spoil the validity of the perturbative

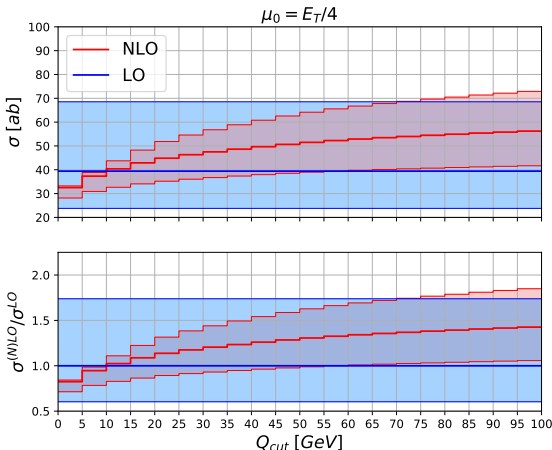

Figure 1: *Integrated fiducial cross-section as a function of $Q_{cut}$ at LO (blue) and NLO (red) in QCD. The bottom panel shows the ratio to the LO predictions. Scale uncertainties, obtained from the 7-point variation, are displayed in both panels. Figure is taken from Ref. [12].*

treatment of the calculation. This occurs because these topologies do not mimic corrections to the LO amplitudes, where the two light jets are always well separated and have an invariant mass equal to the $W$ boson mass. To suppress these effects, we require that at least one pair of light jets has an invariant mass ($M_{jj}$) close to the $W$ boson mass ($m_W$), as specified by the $Q_{cut}$ parameter defined below

$$|M_{jj} - m_W| < Q_{cut}. \tag{1}$$

The default value for this parameter is set to $Q_{cut} = 25$ GeV but we also investigate the effects of varying the $Q_{cut}$ cut throughout our computations.

## 3  Integrated fiducial cross sections

In this section, we present results at the integrated fiducial cross-section level both at LO and NLO in QCD. These results have been obtained using the (N)LO MSHT20 PDF set for the (N)LO calculations and the dynamical scale choice $\mu_0 = E_T/4$, for the renormalization and the factorization scale defined as:

$$E_T = \sum_{i=1,2} \sqrt{m_t^2 + p_T^2(t_i)} + \sum_{i=1,2} \sqrt{m_t^2 + p_T^2(\bar{t}_i)}. \tag{2}$$

We begin by illustrating the dependence of the fiducial cross sections on the $Q_{cut}$ parameter at both LO and NLO in QCD in Figure 1. As expected, the LO cross section is independent of the $Q_{cut}$ parameter, since both light jets have an invariant mass equal to $m_W$ at LO. However, the NLO cross section increases rapidly with higher values of $Q_{cut}$, reaching $\mathcal{K}$ factors as large as 1.45 when $Q_{cut} = 100$ GeV. Additionally, the size of the NLO scale uncertainties, obtained from the usual 7-point variation, also increases for larger $Q_{cut}$ values. This indicates that, for these scenarios, the accuracy of the NLO result is comparable to that of a LO calculation. The fiducial cross sections at NLO for the default case with $Q_{cut} = 25$ GeV, as well as for the extreme case where $Q_{cut} \to \infty$, are presented below:

$$\sigma^{\mathrm{NLO}}_{Q_{cut}=25 \text{ GeV}} = 44.91(2)^{+18\%}_{-23\%}, \qquad \sigma^{\mathrm{NLO}}_{Q_{cut}\to\infty} = 70.19(7)^{+42\%}_{-30\%}. \tag{3}$$

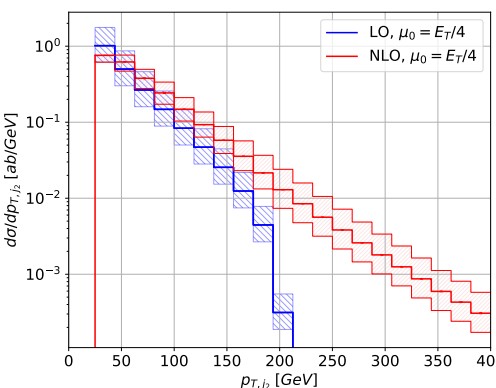 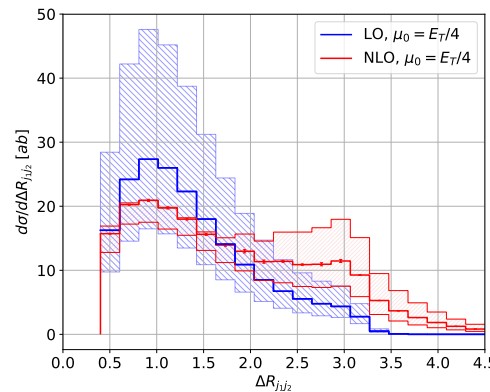

Figure 2: *Differential cross-section distributions for the $pp \to t\bar{t}t\bar{t}$ process in the $3\ell$ channel for the $p_{T,j_2}$ (left) and $\Delta R_{j_1 j_2}$ (right) observables. The (N)LO predictions are shown in (red) blue and uncertainty bands obtained from the 7-point variation are included in both plots. Figures are taken from Ref. [12].*

We notice that the magnitude of the scale uncertainties for our default setup is 23%, while for the case where no restriction is applied on $M_{jj}$, the corresponding uncertainty increases to 42%. Meanwhile, the size of the QCD corrections for the case where $Q_{cut} \to \infty$ are quite substantial, reaching values up to 80%. This highlights the importance of imposing a restriction on $M_{jj}$ to ensure the convergence of the perturbative treatment of the calculation.

Before concluding this section, it is important to note that the various NWA treatments are also highly sensitive to the choice of $Q_{cut}$. The most significant discrepancies were observed in the $Q_{cut} < 15$ GeV regime, particularly between $\text{NLO}_{\text{LO}_{\text{dec}}}$ and $\text{NLO}_{\text{exp}}$, where differences reached up to 40% at $Q_{cut} = 5$ GeV, exceeding the scale uncertainties. At larger $Q_{cut}$ values, around 100 GeV, $\text{NLO}_{\text{full}}$ deviated by up to 20% from the default setup, though these variations were well within the range of the large scale uncertainties. Finally, for our default setup with $Q_{cut} = 25$ GeV, the differences between the various NWA approaches ranged from 10% to 12%, which is well within the size of the theoretical uncertainties. This justifies selecting $Q_{cut} = 25$ GeV as the default value in our computations, as it effectively mitigates significant differences among the various NWA treatments while also preventing large $\mathcal{K}$ factors.

## 4   Differential distributions

To draw meaningful conclusions about how QCD corrections behave in different phase-space regions, it is essential to present results also at the differential level. All the plots in this section have been generated using the defualt value of $Q_{cut} = 25$ GeV. In Figure 2, we display the transverse momentum of the second hardest light jet ($p_{T,j_2}$) and the angular separation between the hardest and the second hardest light jet ($\Delta R_{j_1 j_2}$) at LO (blue) and NLO QCD (red). As shown in Figure 2, there are notable shape distortions between the LO and NLO predictions for both observables. For $p_{T,j_2}$, a sharp cutoff is observed around 200 GeV, which arises purely from the kinematics at LO. The absence of such a restriction at NLO leads to a significant number of events appearing above this value. In the $\Delta R_{j_1 j_2}$ distribution, we notice that at LO, the two light jets are predominantly produced near $\Delta R_{j_1 j_2} \approx 1$. However, at NLO, a second peak appears around $\Delta R_{j_1 j_2} = \pi$, suggesting that the two hardest light jets are produced back-to-back. This behavior indicates that the additional radiation at NLO disrupts the LO kinematical configuration, which favors $\Delta R_{j_1 j_2} = 1$.

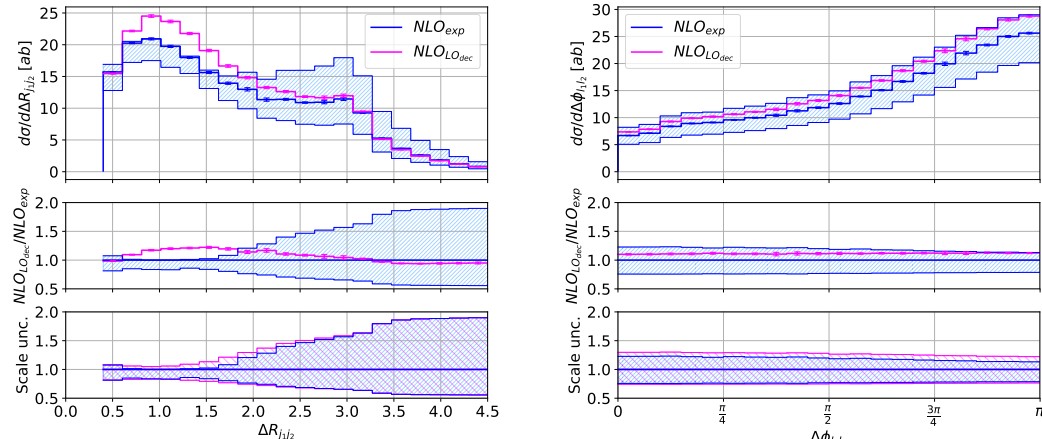

Figure 3: *Differential cross-section distributions for the $pp \rightarrow t\bar{t}t\bar{t}$ process in the $3\ell$ channel for the $\text{NLO}_{\text{exp}}$ (blue) and $\text{NLO}_{\text{LO}_{\text{dec}}}$ (magenta) for the $\Delta R_{j_1 j_2}$ (left) and $\Delta\phi_{l_1 l_2}$ (right) observables. The upper panels display the absolute predictions while in the middle panels the ratio to the $\text{NLO}_{\text{exp}}$ is also provided. In the bottom panels, the NLO scale uncertainties for the two approaches, normalised to their corresponding NLO results are also depicted. Figures are taken from Ref. [12].*

In addition to the magnitude of the QCD corrections, Figure 3 illustrates the impact of the higher-order effects at the decay stage of the top quarks by comparing the $\text{NLO}_{\text{exp}}$ and $\text{NLO}_{\text{LO}_{\text{dec}}}$ approaches. Specifically, we present the angular separation between the two hardest light jets ($\Delta R_{j_1 j_2}$) and the azimuthal angle between the two hardest charged leptons ($\Delta\phi_{l_1 l_2}$), both of which are highly sensitive to spin correlations. The absolute predictions are displayed in the upper panels, while the middle panels show the ratio relative to our default setup. Additionally, the bottom panels depict the size of the scale uncertainties. Omitting QCD corrections during the decay stage of the four top quarks has an impact of up to 22% for the $\Delta R_{j_1 j_2}$ observable, a deviation that lies outside the scale uncertainty range for the $\text{NLO}_{\text{exp}}$ result. In contrast, for $\Delta\phi_{l_1 l_2}$, these effects are smaller, reaching only up to 10%, and are fully covered by the scale uncertainty bands. Finally, from the bottom panels, it is evident that the magnitude of the scale uncertainties is smaller if QCD corrections are applied in both the production and decays of the top quarks.

## 5 Conclusion

In these proceedings, we presented NLO QCD corrections for the $pp \rightarrow t\bar{t}t\bar{t}$ process in the $3\ell$ channel. Our results demonstrated sensitivity to the choice of the $Q_{cut}$ parameter, which was introduced to mitigate large QCD corrections and preserve the validity of the perturbative treatment of the calculation. For the default setup utilising $Q_{cut} = 25$ GeV the impact of QCD corrections was found to be quite important, especially at the differential cross-section level where significant shape distortions were observed. The absence of higher-order QCD effects at the decays of the top quarks was about 10% at the integrated level and up to 22% for some dimensionless observables at the differential level. To further investigate the role of hard emissions in top-quark decays and assess the significance of the NLO spin correlations, a comparison with NLO QCD results matched to parton showers is essential. This approach will also allow us to evaluate the impact of Matrix Element Corrections during the showering

process, as discussed, for example, in Ref. [17, 18]. In a future work, we intend to conduct such a comparison for this decay channel.

## Acknowledgements

This work was supported by the German Research Foundation (Deutsche Forschungsgemeinschaft - DFG) under grant 396021762 - TRR 257: Particle Physics Phenomenology after the Higgs Discovery, and grant 400140256 - GRK 2497: The Physics of the Heaviest Particles at the LHC.

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
