# Peer review of "$t\bar{t}t\bar{t}$: NLO QCD corrections in production and decays for the $3\ell$ channel"

_SciPost Physics Proceedings_

## Round 1 · Referee Report · Anonymous (Referee 1) · 2025-1-24

Report

The author investigates NLO QCD corrections to four-top-quark production, focusing on the novel inclusion of QCD corrections in the decay processes using the NWA. The results reveal that the corrections in the decay are appreciable and lead to reduced perturbative uncertainties, providing a more precise prediction for this very complex process.

Requested changes

The author notes that the spin correlations, that are implemented using the method described in Ref. [hep-ph/0702198], are accurate at LO. However, Ref. [hep-ph/0702198] claims NLO accuracy "for hard real emissions". I kindly request the author to clarify this apparent discrepancy in the manuscript.

Recommendation

Ask for minor revision

  • validity: high
  • significance: high
  • originality: -
  • clarity: high
  • formatting: excellent
  • grammar: excellent

Author:  Nikolaos Dimitrakopoulos  on 2025-01-27  [id 5152]

(in reply to Report 1 on 2025-01-24)

I would like to thank the reviewer for taking the time and the effort to provide this feedback. In my resubmission I have made the following changes in order to clarify the discrepancy mentioned in the report:

In the Introduction, the sentence "As a result, spin correlations were retained at LO accuracy and QCD corrections to the top-quark decays were only described by the parton shower" was changed to "As a result, spin correlations were retained at LO accuracy in the soft and collinear regions and at NLO accuracy only for hard real emissions. Furthermore, QCD corrections to the top-quark decays were only described by the parton shower."

Also, again in the Introduction the sentence "By doing so, the emission of the hard radiation was well described in the top-quark decays and spin correlations were preserved to NLO accuracy" was changed to "By doing so, the emission of the hard radiation was well described in the top-quark decays and spin correlations were preserved to NLO accuracy in the whole phase space without any approximation".

These updates should now address the discrepancy in spin-correlation accuracy between Ref. [hep-ph/0702198] and my work.

Greetings,
Nikolaos Dimitrakopoulos

---

## Editorial Decision

editorial_decision: